# Carbon Nanotube Devices for Quantum Technology

**DOI:** 10.3390/ma15041535

**Published:** 2022-02-18

**Authors:** Andrey Baydin, Fuyang Tay, Jichao Fan, Manukumara Manjappa, Weilu Gao, Junichiro Kono

**Affiliations:** 1Department of Electrical and Computer Engineering, Rice University, Houston, TX 77005, USA; ft13@rice.edu (F.T.); mmanjappa@rice.edu (M.M.); 2Smalley-Curl Institute, Rice University, Houston, TX 77005, USA; 3Applied Physics Graduate Program, Smalley-Curl Institute, Rice University, Houston, TX 77005, USA; 4Department of Electrical and Computer Engineering, University of Utah, Salt Lake City, UT 84112, USA; jichao.fan@utah.edu (J.F.); weilu.gao@utah.edu (W.G.); 5Department of Physics and Astronomy, Rice University, Houston, TX 77005, USA; 6Department of Materials Science and NanoEngineering, Rice University, Houston, TX 77005, USA

**Keywords:** carbon nanotubes, quantum technology

## Abstract

Carbon nanotubes, quintessentially one-dimensional quantum objects, possess a variety of electrical, optical, and mechanical properties that are suited for developing devices that operate on quantum mechanical principles. The states of one-dimensional electrons, excitons, and phonons in carbon nanotubes with exceptionally large quantization energies are promising for high-operating-temperature quantum devices. Here, we discuss recent progress in the development of carbon-nanotube-based devices for quantum technology, i.e., quantum mechanical strategies for revolutionizing computation, sensing, and communication. We cover fundamental properties of carbon nanotubes, their growth and purification methods, and methodologies for assembling them into architectures of ordered nanotubes that manifest macroscopic quantum properties. Most importantly, recent developments and proposals for quantum information processing devices based on individual and assembled nanotubes are reviewed.

## 1. Introduction

Quantum technology, which includes quantum sensing, imaging, computing, communication, and simulation, relies on quantum mechanical principles to radically change the way we store, process, and transmit information. Quantum devices are being developed by many research groups using various material platforms. Some examples include superconducting circuits for applications in quantum computing [1,2], cold trapped ions used for quantum simulations [3,4,5], and spins of nitrogen vacancy (NV) centers in diamond as ultrasensitive magnetometers [6,7]. For quantum communications, many solid-state single photon emitters have been demonstrated [8], including functionalized carbon nanotubes (CNTs).

CNTs are quintessentially one-dimensional (1D) quantum objects, possessing a variety of extraordinary electrical, optical, and mechanical properties [9,10,11,12,13,14,15]. These properties are ideally suited for developing devices that operate on quantum mechanical principles, especially for quantum information processing (QIP). The states of 1D electrons, excitons, and phonons in CNTs with exceptionally large quantization energies are promising for high-operating-temperature quantum devices. For example, CNTs have been demonstrated to work as room-temperature single photon emitters at telecommunication frequencies with deterministic placement of emitters and polarization control [16,17,18,19,20] (see details given in Section 3.2).

In this review, we summarize research progress in the development of CNT-based quantum devices. Recent developments in preparing macroscopic films of highly ordered, or aligned CNTs in film or fiber form [21,22,23,24,25,26,27,28] have enabled fundamental studies of 1D physics as well as development of unique 1D devices on a macroscopic scale [13,14]. For example, strong optical anisotropy has resulted in CNT-based polarizers [21,29,30,31], polarization rotators [32], and modulators [33,34,35,36]. Metallic and doped aligned CNT films are natural hyperbolic materials [37,38,39], in which the epsilon near point is located in the infrared and can be tubed by doping [40]. Due to the large Seebeck coefficients of CNTs [41,42,43], several ultra-broadband thermoelectric photodetectors have been successfully demonstrated [41,44,45,46].

This review article is organized as follows. First, in Section 2, we introduce CNTs and describe their basic properties. Then, we discuss CNT devices for quantum technology, based on individual CNTs (Section 3) and assembled CNTs (Section 4). While a number of previous reports focused on different aspects of quantum information processing (QIP) such as CNT-based quantum entanglement [47,48], hybrid superconducting qubits that employ a CNT [49], nanoassembly techniques of CNTs for hybrid circuit quantum electrodynamics (QED) devices [50], and transmission line filters [51], we limit ourselves to charge, spin, and phonon CNT qubits, single photon emitters, and exciton polaritons. Currently available state-of-the-art techniques for fabricating macroscopic films of aligned CNTs are also reviewed.

## 2. Basics of Carbon Nanotubes

There are four members in the $sp^{2}$ family of carbon allotropes with different dimensions: C_60_ (0D), carbon nanotubes (1D), graphene (2D), and graphite (3D). In any of these materials, each carbon atom is covalently bonded with three adjacent carbon atoms in a honeycomb lattice. Their atomic structures are closely related, where graphene can be viewed as a fundamental building block [52]. For example, and particularly relevant to this article, a single-wall CNT (SWCNT) can be viewed as a rolled-up version of monolayer graphene.

Figure 1a shows a graphene lattice, which has two primitive lattice vectors $a_{1}$ and $a_{2}$ in real space. The chiral (or roll-up) vector $C=na_{1}+ma_{2}$ defines the rolling-up direction of graphene, where *n* and *m* are two positive integers with m≤n, called chirality indices. Two highly symmetric rolling manners of the graphene sheet lead to two types of achiral SWCNTs, known as zigzag SWCNTs (Figure 1b) with chirality indices (m,0) and armchair SWCNTs (Figure 1c) with equal chirality indices (m,m). All other SWCNTs are generally referred to as chiral SWCNTs.

The electronic band structure of SWCNTs sensitively depends on (*n*,*m*), and the quantity ν=(n−m) mod 3 classifies SWCNTs into three distinct categories [53,54,55]. Specifically, SWCNTs with ν=0 and n=m (armchair SWCNTs) are metallic; SWCNTs with ν=0 but n≠m possess small curvature-induced band gaps; and SWCNTs with ν=1 or 2 are semiconductors with a band gap spanning from the visible to the near-infrared ranges. Figure 1d,e schematically shows the band dispersions (left) and density of states (DOS) (right) of metallic and semiconducting SWCNTs, respectively. The peaks occurring in the DOS are 1D van Hove singularities. Such an extremely concentrated DOS leads to significant light absorption and emission through the generation and recombination of electron–hole pairs or excitons. Due to reduced screening of the Coulomb interactions characteristic of low-dimensional systems, exciton binding energies in SWCNTs are significantly larger than those in traditional semiconductors. For example, GaAs and its quantum-well structures possess exciton binding energies of the order of ∼10 meV [56], whereas those in SWCNTs can be hundreds of meV [57,58].

Extremely anisotropic electronic and excitonic states in SWCNTs lead to distinct polarization-dependent optical selection rules [59,60,61,62,63,64]. Optical attenuation spectra for light polarization parallel and perpendicular to the nanotube axis in an aligned SWCNT film are shown in Figure 2a,b. For parallel polarization, there is a broad peak in the mid-infrared range arising from the longitudinal plasmon resonance [65,66], while no absorption is detected for perpendicular polarization below ∼1 eV. Figure 2b shows attenuation only in the near-infrared and visible ranges, showing excitonic interband absorption peaks [21]. Figure 2c describes the allowed lowest-energy interband excitonic transitions in a semiconducting SWCNT [63]. Arrows of different colors show allowed optical transitions for light polarized along the tube axis (red) and perpendicular to the tube axis (blue). Each conduction or valence subband has a well-defined angular momentum with quantum number *i* and an eigenvalue of the Pauli matrix $\sigma_{x}$ [9,67]. Light with polarization parallel to the nanotube axis can induce transitions with Δi=0. For light polarized perpendicular to the nanotube axis, the following selection rules must be satisfied: (i) Δi=±1, meaning that the angular momentum quantum number has to change by ±1 and (ii) $\Delta \sigma_{x}=0$, meaning that the eigenvalue of the Pauli matrix $\sigma_{x}$ has to be conserved. Thus, in Figure 2b, the first two interband transitions for semiconducting nanotubes ($E_{11}^{S}$ and $E_{22}^{S}$) and the first interband transition in metallic nanotubes ($E_{11}^{M}$) are observed. For perpendicular polarization, these peaks are completely absent, and a broad absorption feature is instead observed due to cross-polarized excitons, whose intensity is suppressed due to a depolarization effect [21,60,63].

Similar to the unique electronic properties, SWCNTs also host unique phononic properties. These include the radial breathing mode (RBM), the twist acoustic mode, and G modes (see Figure 3a). For example, RBM has been used to probe the structure of SWCNTs as its frequency is simply inversely proportional to the diameter of an isolated nanotube [68]. In order to access and study phonons in carbon nanotubes, Raman spectroscopy has been widely utilized [69]. Because of the existence of the van Hove singularities in carbon nanotubes, the vibrational properties of individual nanotubes can be measured through resonantly enhanced Raman scattering [70]. For more details about Raman spectroscopy in CNTs, we refer the reader to the literature [69,71,72].

Another useful technique for studying phonons in CNTs is coherent phonon (CP) spectroscopy. In general, CP spectroscopy is advantageous for detecting phonon modes with frequencies smaller than 100 cm${}^{-1}$, due to the absence of a Rayleigh scattering background. CP spectroscopy is an ultrafast pump–probe spectroscopy technique where the transient reflection or transmission signal with respect to the delay time between pump and probe pulses exhibits oscillations corresponding to phonon frequencies [73] (see Figure 3b,c). The first observation of CP oscillations in SWCNTs was reported by Lim et al. [74]. The phonon frequencies in CP spectroscopy are obtained by Fourier transforming the time-domain signal. Analysis of these experiments provides a wealth of information on the chirality dependence of light absorption, phonon generation, and phonon-induced band structure modulations [73,74,75,76,77,78].

Different CNT properties need to be considered for different types of quantum device applications. For CNT quantum dots, most of the physics is a consequence of the one-dimensional confinement of electrons occurring on a hexagonal lattice. Therefore, the chirality of nanotubes is not essential, but narrow-gap nanotubes (non-armchair mod-3 nanotubes) are preferred because their Fermi energy can be tuned easily by electrostatic gating [79]. In general, the less defective the tubes, the better for quantum device applications. Specifically, for spin-based CNT quantum devices, nanotubes grown from ^12^C ions are preferred to avoid electron spin decoherence caused by nuclear spins. For photonic quantum devices, it is critical to have nanotubes with the desired band gaps for different applications. However, overall, systematic chirality-dependent studies are lacking for quantum devices, as it is still a challenge to produce chirality-separated CNTs in large quantities. Therefore, state-of-the-art quantum technology applications of CNTs are advancing together with advancements in sample purification and preparation.

## 3. Devices Based on Individual Carbon Nanotubes

In this section, we review the development of quantum devices based on a single CNT with defined (*n*,*m*), exploiting the quantum mechanical properties at the single-electron and single-photon levels. Gating a single CNT can produce a spin-, charge-, or phonon-based qubit possessing a long coherence time and a high-quality-factor resonance, which is promising for sensitive magnetometry, quantum gates, and optomechanical nanocircuits. Furthermore, chemical functionalization of SWCNTs assists the formation of spin qubits and room-temperature single-photon emitters for quantum information processing in the telecommunication band.

### 3.1. Single-Electron Devices

Here, we discuss some encouraging developments made in recent years on SWCNT-based single-electron devices, nanocircuits, and hybrid quantum devices coupled to NV centers in diamond, which are especially promising for controlled and tunable quantum information processing applications.

#### 3.1.1. Charge and Spin Qubits

A CNT quantum dot (QD), or an artificial atom, is formed when a single CNT is suspended between two electrical contacts. In such a situation, electrons are confined in a potential well with discrete energy levels. Electrons can be added into these energy levels one by one. CNT-based QDs have been demonstrated to be promising for QIP applications, either as charge or spin qubits [80]. The advantage of using CNT QDs as spin qubits is that long spin relaxation times are expected because of the small spin–orbit interaction and nonexistent nuclear spin interaction in CNTs, while for charge qubits, the energy level spacing is larger (submillimeter to the THz range) [81] than that of traditional semiconductor-based charge qubits [80].

Different types of charge-based CNT qubits have been demonstrated [82,83,84,85,86]. Figure 4 shows an atomic force microscopy (AFM) image of a CNT double QD device, where quantum dots can be created by tuning the voltage on the top gates (TG). The side gates (SG) allow each QD to be addressed individually [87]. Noninvasive readout techniques such as a combination of radio-frequency reflectometry of the quantum capacitance with microwave manipulation of charge qubit states can be used to operate the QD device at an optimal point where charge coherence is extended [85]. More recently, an atom-like charge qubit in a CNT based on a single QD has been shown to have high sensitivity to electric and magnetic fields. The electric field sensitivity was better than that of a single-electron transistor, and the DC magnetic field sensitivity was comparable to that of NV centers [86].

As well as charge qubits, spin qubits have also been reported [79,83,84,88,89,90]. For instance, a valley–spin qubit in a CNT has been demonstrated with a coherence time of about 65 ns [83]. In another study, Pei et al. achieved longer coherence times, limited by charge noise in the device [89]. Most recently, Cubaynes et al. coupled a single electronic spin in a CNT-based double QD with a microwave cavity. They showed a tunable and coherent spin–photon interface with a coupling strength of ∼2π×2.0 MHz and a low decoherence rate of ∼2π×250 kHz, which is suitable for future swap experiments. Figure 4d–f summarizes a nuclear-spin-limited spin–photon interface of the qubit. The linewidth and derivative of the dispersion relation of the spin transition, spin–photon coupling strength, and cooperativity as a function of detuning are shown in Figure 4d–f, respectively [90]. Another approach to spin-based qubits is chemical functionalization of SWCNTs, which results in localized spins at the defect sites [91].

#### 3.1.2. Nanomechanical Qubits

The exceptional mechanical properties of CNTs make them one of the materials of choice for nano-electromechanical systems (NEMS). More specifically, CNTs offer ultrasmall diameters and therefore ultrahigh aspect ratios and ultralow mass. Most of the CNT NEMS devices are resonators where the nanotube serves as the resonator body. Several key developments such as tunability of the resonance frequency [92], high quality factors [93,94], and operation in the GHz frequency range [95,96] have been demonstrated. Thus, CNT-based resonators can lead to the development of ultrasensitive mass and force detectors, as well as quantum bits with long coherence times based entirely on mechanical motion [97].

There have been several proposals to realize nanomechanical qubits. Rips and Hartmann envisioned a device that consists of an array of doubly clamped nanobeams coupling to a common resonance mode of a high-finesse optical cavity. Experimentally, such a setup can be achieved with CNTs and an evanescent field of a whispering-gallery-mode cavity. Small tip electrodes can be used to manipulate each nanobeam individually with electrostatic and radio-frequency fields. The key is the introduction of anharmonicity to the mechanical spectrum via applied electric fields, which allows coupling of the cavity mode to the ground and first excited states of the nanobeam [98,99].

A more recent proposal focuses on the coupling of the flexural modes of a CNT to an integrated double quantum dot, with the dot itself defined in the nanotube [100]. Figure 5a,b shows schematic diagrams of the proposed nanomechanical qubit. By using multiple gates, a double well potential can be engineered. Only two states, each with one excess electron, are considered. They are coupled by a hopping term t/2 and have an energy difference ϵ (see Figure 5a), which is controlled by two gate voltages. The double dot is placed in the center of the nanotube, to efficiently couple the two charge states with the second (antisymmetric) mechanical mode. Such a qubit, which is mainly of a mechanical character, offers very long coherence times. Moreover, the nanomechanical qubits can be coupled to each other by microwave cavities, which allows the implementation of qubit operations such as controlled-NOT gates using only microwaves.

#### 3.1.3. Spin–Nanomechanical Hybrid Devices

In this section, we review recent experiments and theoretical proposals related to coupling between spins and phonons in CNTs for hybrid quantum information devices. While single spins can serve as qubits, coupling to mechanical motion of CNTs can enable detection and manipulation of spin qubits [101]. To achieve strong coupling between spins and phonons, CNTs can be either functionalized with magnetic ions [102] or molecules [103], or interfaced with solid-state qubits such as NV centers in diamond [104].

A strong spin–phonon coupling between a single molecular spin based on a pyrene-substituted bis(phthalocyaninato)terbium(III) (TbPc${}_{2}$) and the quantized mechanical motion of a CNT resonator has been demonstrated [103]. In another study, Gd${}^{3+}$ ions were used to fill double-wall CNTs (DWCNTs) and spin–phonon coupling was investigated with low-temperature Raman spectroscopy. The spin–phonon coupling strength was shown to be about three times higher than that for other multiferroic compounds [102]. Mechanically interlocked Cu${}^{2+}$ and Co${}^{2+}$ metalloporphyrin dimer rings around a SWCNT, or a mechanically interlocked nanotube (MINT), have also been demonstrated, as shown in the transmission electron microscope (TEM) image in Figure 6a. For such mechanical bonds, the porphyrin magnetic cores are brought close to a CNT without disturbing the molecular spin or the CNT. Quantum coherence times of the molecular spins were not altered by the formation of MINTs. In addition, as noted by the authors, such molecular hybrids can be placed into nanoscale devices deterministically using dielectrophoresis [105].

Solid-state-based spin qubits, and especially negatively charged NV centers in diamond, have been thoroughly investigated and are already being used as magnetometers for the exploration of condensed matter physics [106]. NV centers can be manipulated and entangled at temperatures well above room temperature [107,108], and thus are one of the systems of choice for quantum information science and technology. To date, there have been several proposals for experimental schemes to realize coupling between NV centers and CNTs, and NV-center entanglement via CNT. Li et al. proposed and theoretically demonstrated strong coupling of an NV center and a DC-current-carrying CNT placed nearby. They also showed dynamic control over the magnetomechanical interaction, which can be achieved using external driving microwave fields and an electric current through the nanotube. A schematic of the proposed CNT–NV hybrid device is shown in Figure 6b. Two main advantages of this approach are the intrinsic nature of the coupling, i.e., no external magnetic tips are required to tune the coupling strength and the scalability of the NV centers array in diamond [104].

Most recently, several reports have proposed schemes for macroscopic entanglement of NV centers in diamond via a CNT [109,110,111,112]. For example, Ma et al. considered the coupling of a two-NV-centers ensemble in diamond to the phonon mode of a CNT. Macroscopic entanglement was studied in the presence of spin dephasing, relaxation, and mechanical dissipation. Control of entanglement of NV centers by tuning the distance from the diamond nanocrystal, the dimensions of the nanotube, the microwave field, and the current has also been shown [110]. Dong et al. designed a scalable architecture for a hybrid spin–mechanical quantum entanglement system based on NV centers and CNTs, as depicted in Figure 6c, showing an NV center array under DC-current-carrying CNTs [109].

### 3.2. Single-Photon Devices

Like single-electron systems, single-photon devices also play a key role in enabling new-generation quantum solutions and technologies. In recent years, quantum-defect engineered solid-state materials have enabled stable room-temperature single-photon emission [8] for a wide range of applications, including secure quantum information and communication, precision measurements, and high-resolution imaging. Although a variety of materials are being investigated for single-photon emission across a wide energy spectrum, a major challenge exists in meeting the criteria for room temperature and fully integrable, bright single-photon sources operating at telecommunication wavelengths for energy-efficient quantum communication protocols. Recent studies on quantum defect engineering of photoluminescence (PL) emission in semiconducting SWCNTs [20,113,114] for tuning PL intensity and wavelengths across the near-infrared regime (850 nm to 2 μm) have enriched the prospects of SWCNTs as one of the most promising materials for realizing scalable and energy-efficient quantum solutions in the technologically significant telecommunication bands.

Here, we review the advancements of semiconducting SWCNTs-based single-photon emitters, starting from initial studies concerning exciton localization at cryogenic temperatures [115,116,117] to the recent progress in chemical-defect-engineered deep trap states (two-level system) with energy much larger than the thermal energy ($k_{B}T$ = 26 meV, where $k_{B}$ is the Boltzmann constant and *T* is the temperature) at room temperature, signifying the capability for single-photon emission at room temperature. Alongside this, nanotube chirality (n,m) and diameter-dependent tunability of emission wavelengths enables tunable single-photon emission across the telecommunication wavelengths. Furthermore, studies concerning photon indistinguishability [19] and enhancement of single-photon emission rates are discussed in the realm of cavity QED.

#### 3.2.1. Single-Photon Emitters

In undoped SWCNTs, tightly bound excitons freely diffuse along the tube length, eventually producing band-edge E${}_{11}$ emission in the near-infrared. Nonclassical behavior in the E${}_{11}$ PL emission from an undoped semiconducting SWCNT was first reported by Högele et al. [115] for a (6,4) nanotube at around 860 nm at low temperatures. Hanbury Brown–Twiss (HBT) measurements demonstrated single-photon emission or photon antibunching, i.e., strong suppression in the two-photon intensity correlation $g^{\left(2\right)}$(0) reaching as low as 3%. The authors proposed that the observed photon antibunching behavior is due to exciton localization at low temperatures, as evidenced by the reduced linewidth of the PL peak at temperatures below 10 K (see Figure 7a). Pump-power-dependent studies on the PL lifetime revealed an exciton–exciton annihilation process contributing to the quantum emission through the inhibition of simultaneous two-photon emission events. It was also shown that with increasing pump intensity, the exciton emission intensity saturates, along with fluctuations in the PL intensity and the wavelength that is shown to be sensitive to the temperature cycling, which is a characteristic signature of a quantum emitter.

Further strengthening of exciton localization for realizing bright and long-lived quantum emission has been demonstrated by Hoffmann et al. by using suspended SWCNT quantum dots at low temperatures [116]. The authors studied the PL emission properties of CVD-grown suspended quantum dot nanotubes that showed remarkably narrow emission lines (lifetimes ∼ 3.35 ns) centered at 1.36 eV, compared to a broad emission line for CoMoCAT CNTs in contact with SiO${}_{2}$. Photon correlation measurements revealed strong antibunching for the emission light from the suspended CNT quantum dots, achieving $g^{\left(2\right)}$(0) = 0.3, whereas photon bunching was observed for CVD-grown CNTs in contact with the SiO${}_{2}$ substrate. Localized excitons in the suspended CNT quantum dots exhibited intrinsically bright, long-lived, and coherent quantum emission at low temperatures, due to suppression of PL quenching.

Room-temperature single-photon emission from semiconducting SWCNTs was first demonstrated by Ma et al. [118] through chemical functionalization of undoped (6,5) SWCNTs, operating in the 1100–1300 nm wavelength range. Solitary oxygen defects were created through encapsulation of a SiO${}_{2}$ layer on surfactant-wrapped, undoped (6,5) SWCNTs, which resulted in ether-d or epoxide-l functional groups creating deep trap states $E_{11}^{*}$ and $E_{11}^{*-}$ located at 130 meV and 300 meV below the $E_{11}$ band-edge emission, respectively (see Figure 7b). These dopant-induced deep trap states of SWCNTs resembling a quantum two-level system are responsible for single-photon emission at room temperature. Their photon correlation (HBT) measurements on $E_{11}^{*}$ and $E_{11}^{*-}$ at 1108 nm and 1272 nm, respectively, revealed nearly vanishing $g^{\left(2\right)}\left(0\right)$ (∼0.05 and 0.32) at 150 K and 298 K. Such quantum-defect-enabled SWCNT single-photon emitters have also been used for performing high-resolution and high-contrast in vivo imaging of deep tissue, while greatly suppressing autofluorescence backgrounds [119].

He et al. were able to tune the quantum emission wavelengths towards the O and C telecom bands through covalently introduced $sp^{3}$ defects in nanotubes of different chirality [17]. They achieved a nearly continuous and wide tunability for single-photon emission between 1100 nm and 1600 nm by engineering defect states in (6,5), (7,5), and (10,3) SWCNTs, observing longer-wavelength emission as the tube diameter increased, as shown in Figure 7c. Aryl $sp^{3}$ defects were introduced through covalent functionalization of nanotubes using diazonium compounds to create deep trap states. The covalently bound $sp^{3}$ defect sites minimize PL blinking in the functionalized nanotubes through reduced perturbation in the nanotube electrostatic environment, thereby achieving ultrahigh-stability single-photon emission with an emission purity of nearly 99% at room temperature.

Precisely controlled positioning of defects in the host material has been a strong focus in engineering quantum light emitters for both quantum network applications and fundamental studies of photon correlation physics. Recently, patterning of defect sites has been demonstrated through covalent DNA functionalization of nanotubes, which locally modulates the energy levels and emission wavelengths, depending on the nucleotide sequence used to coat the nanotube [120]. The desired functionalization was achieved by exposing the single-stranded DNA-coated SWCNT in an aqueous solution to a single oxygen ${}^{1}\Delta_{g}$ under ambient conditions, forming selective covalent bonding between the nanotubes and the guanine nucleotides, as shown by the red dot in Figure 7d. These guanine-functionalized nanotubes constituted high densities of relatively shallow exciton trap sites (with axial separations of less than 1 nm along the nanotube length), forming a composite defect with a deeper trap state and resulting in an overall redshift of the exciton emission peak.

More recently, HBT photon correlation measurements revealed single-photon emission from guanine functionalized (6,5) nanotubes with emission purity of nearly 8% at 4 K and 27% at room temperature [16]. The auto- and cross-correlation measurements on closely spaced quantum emitters in a single nanotube revealed $g_{22}^{\left(2\right)}\left(0\right)=0.07$ and $g_{12}^{\left(2\right)}\left(0\right)=0.5$, respectively, showing weak interactions between individual emitters. The authors proposed that the cross correlations between emitters are established via the capture processes of the band-edge exciton with the coupling strengths determined by the shelving and deshelving rates given by $\kappa_{s}$ and $\kappa_{d}$, respectively. This study provided a promising approach toward realizing patterned single-photon emitters for novel investigations on photon–photon interactions and coherence-mediated effects in single- and multi-photon quantum emission regimes.

Electrically driven SWCNTs exhibiting electroluminescence (EL) have also been shown to emit single photons at cryogenic temperatures [18]. An array of sorted semiconducting SWCNTs was placed in a nanophotonic circuit consisting of a waveguide and connected to three pairs of gold electrodes to give a purely electrical excitation under a strong DC bias. The HBT configuration for photon autocorrelation measurements was constructed on chip by symmetrically placing superconducting nanowire single-photon detectors (SNSPD). This study revealed photon antibunching in the EL at 1370 nm, showing a minimum value of $g^{\left(2\right)}\left(0\right)$ = 0.49. This approach revealed an attractive feature of EL SWCNT single-photon sources to enable on-chip and fully electrically triggered nanophotonic circuits for integrated quantum networks.

With the unique capabilities of producing single photons either electrically or optically and the ease of integration with photonic and plasmonics platforms, SWCNT-based single-photon devices are promising for establishing efficient and on-chip quantum emitters in emerging hybrid and integrated quantum photonic technologies. In the next section, we review some of the latest developments in further enhancing single-photon emission from individual SWCNT-based devices through cavities.

#### 3.2.2. Cavity Enhancement of Single-Photon Emission

Implementing electromagnetic cavities opens a new avenue for controlling the light–matter coupling strength, *g*. Multiple light–matter coupling regimes exist, including the weak coupling (WC), strong coupling (SC), and ultrastrong coupling (USC) regimes. The system is in the WC regime when *g* is smaller than the losses in the system. In this regime, the spontaneous emission rate can be enhanced through the Purcell effect. By contrast, in the SC regime, *g* is higher than all losses in the system; the energy is exchanged between the matter and light periodically at a rate faster than any decay rate, exhibiting coherent Rabi oscillations in the time domain. The USC regime arises when *g* becomes a significant fraction of the bare uncoupled matter and light frequency at zero detuning, $\omega_{0}$, conventionally defined as $g≳0.1\omega_{0}$. Nontrivial vacuum electromagnetic fluctuations can interact with the matter and modify the properties of the system in the USC regime [121]. In this subsection, we review the enhancement of single-photon emission in CNT systems in the WC regime using different cavity designs. Notably, Purcell factors up to 415 have been achieved in CNT color centers [19]. Exciton polaritons in macroscopic CNT devices in the SC and USC regimes are discussed in Section 4.

The photon emission rate can be increased by a cavity in two ways [122]. First, according to the Purcell effect, the spontaneous emission lifetime becomes shorter when the matter is in resonance with the cavity mode, leading to a higher emission intensity. Second, the cavity can confine the photon emission to a specific direction, thus enhancing the intensity of photons reaching the detector. The emission enhancement and linewidth reduction of CNT systems were first studied in a Fabry–Pérot planar cavity consisting of metallic mirrors [123,124,125] and Bragg mirrors [125,126]. Nevertheless, the cavity mode frequency of the planar cavity is determined by the cavity length, which is difficult to tune precisely. The first CNT system with an emission peak exactly matching the cavity mode was reported in 2013 [122]. To circumvent the difficulties in fabrication and enhance the coupling by reducing the mode volume, different cavities have been subsequently used to couple with CNT emitters, such as photonic crystal nanocavities [127,128,129], microdisk resonators [130], microring resonators [131], and plasmonic nanocavities [132].

Walden-Newman et al. observed enhanced single-photon emission from individual SWCNTs inside a cavity [124]. As schematically shown in Figure 8a, SWCNTs were deposited on a polymer/gold/SiO${}_{2}$ structure, which was capped with another polymer layer. The polymer layers acted as a cavity dielectric and the gold layer acted as a mirror to enhance light extraction in the far field. A 50-fold enhancement of exciton emission was observed with the cavity-embedded SWCNT system. The $g^{\left(2\right)}\left(\tau \right)$ measurements under resonant E${}_{22}$ excitation demonstrated single-photon antibunching with $g^{\left(2\right)}\left(0\right)=0.15$.

To ensure spectral and spatial matching between randomly deposited SWCNTs and a cavity, Jeantet et al. proposed a new approach where the top mirror of a microcavity was placed at the apex of a flexible optical fiber [133]. Figure 8b shows a schematic diagram of the emitter and the cavity. An SWCNT layer was first fabricated on a dielectric mirror. Micro-PL measurements were performed to locate a nanotube; an aspherical lens was used in this stage to collect the luminescence. The cavity was then formed by inserting an optical fiber through a hole at the center of the aspherical lens. As the position of other components was not changed, this guaranteed the spatial matching between the nanotube and the cavity. In addition, the cavity frequency could be tuned by changing the cavity length. The Purcell factor of this system was found to be up to 5, and antibunching was observed $\left(g^{\left(2\right)}\left(0\right)∼0.03\right)$.

A substantial enhancement of the Purcell factor was later demonstrated by Luo et al. with a plasmonic nanocavity [132] (see Figure 8c). The electromagnetic mode in the plasmonic nanocavity with a nanometer-scale gap featured ultrasmall mode volumes, leading to a strong enhancement of photon emission. Although plasmonic structures are generally lossy, it was shown that the radiative decay rate of the plasmonic nanocavity, $\gamma_{R}$, can be much greater than the nonradiative decay rate of the system, $\gamma_{\mathrm{NR}}$, when the emitters are placed near (∼2 nm) the cavity mode [135]. A 2 nm thick Al_2_O_3_ spacer layer was grown on the bowtie antenna array to provide the required distance. A 100 nm thick gold mirror was engineered below the SiO${}_{2}$ substrate to amplify the light collection efficiency. The highest Purcell factor achieved in this system was 180, while the average was 57. The cavity-enhanced quantum yield, $\eta =\gamma_{R}/\left(\gamma_{R}+\gamma_{\mathrm{NR}}\right)$, reached 62% (average 42%). The authors also demonstrated antibunching ($g^{\left(2\right)}\left(0\right)=0.30\pm 0.06$).

Electromagnetic cavities were also implemented to investigate the enhancement of single-photon emission from dopant states in CNTs [134]. Doped SWCNTs were deposited on a two-dimensional photonic crystal microcavity, as illustrated in Figure 8d. A Purcell factor of 18 and a cavity-enhanced radiative quantum efficiency of 31% were achieved. Although the SWCNT density in the device was not as low as the individual SWCNT level, clear photon antibunching was still observed ($g^{\left(2\right)}\left(0\right)=0.136\pm 0.005$). This was attributed to cavity coupling and spectral filtering. Note that all measurements were conducted at room temperature.

The performance of doped SWCNT devices was further improved by Luo et al. through integration with an array of plasmonic gap-mode nanocavities [19]. Figure 8e shows doped SWCNTs coupled to plasmonic gap-mode nanocavities. A plasmonic gap mode with a ∼5 nm thick gap was formed between the gold nanocube array and a gold layer. It was shown that a high quantum yield can be achieved when the gap size was at an ultrasmall length scale (2–5 nm) [136,137,138]. The (6,5) SWCNTs were placed at the center to couple with the gap mode. The Purcell factor of this cavity was 415, with η up to 74%. A remarkable single-photon purity was demonstrated: $g^{\left(2\right)}\left(0\right)=0.01\pm 0.005$. In addition, when the device was cooled to 4 K to suppress exciton–phonon dephasing, the coherence time $T_{2}$ increased and became comparable to the spontaneous emission lifetime $T_{1}$, which is shown by the time-resolved PL for coupled and uncoupled SWCNTs in Figure 8f. Under these conditions, quantum decoherence diminishes, and indistinguishable single photons are generated. This work showed a high two-photon interference visibility up to 0.79 using a Hong–Ou–Mandel interferometer [19].

## 4. Devices Based on Assembled Carbon Nanotubes

We have so far looked at quantum devices that are based on single nanotubes. However, to take advantage of the extraordinary properties of individual CNTs for real-world device applications, wafer-scale *aligned* ensembles of *chirality-enriched* CNTs are most desirable. The issue of scalability is self-explanatory. For instance, single-electron or -photon CNT quantum devices eventually need to be produced in large quantities with a controlled spacing or pitch between the nanotubes. Alignment of nanotubes with respect to each other is another necessary requirement to preserve the 1D properties of CNTs on a macroscopic scale, e.g., single-photon emission with the same polarization for all emitters. For polaritonic applications, both the density and thickness of a CNT film are important as the coupling strength between light and matter depends on the interaction volume and the number of dipoles. In this section, we first briefly mention techniques for the separation of different types of nanotubes and preparation of chirality-enriched suspensions. Next, we discuss devices based on both random and aligned CNT films, including an overview of state-of-the-art alignment methods.

For chirality separation, direct-growth methods achieve chirality enrichment during the process of SWCNT synthesis or growth, typically in a chemical vapor deposition process. Both carbon-based templates and SWCNT fragments can be used as growth "seeds" to fabricate SWCNTs along pre-defined structures [139]. Pre-treatment with microwaves can produce micrometer-scale SWCNTs [140]. Additionally, controlling the polarization of the catalyst charge by applying an external electric field to produce almost defect-free SWCNT films is another method for fabricating chirality-enriched crystalline SWCNT film devices, resulting in packing densities of about 50 CNTs per μm [141]. Post-processing methods include several purification techniques, whose aim is to separate one type of specific chirality SWCNTs out of a mixture of SWCNTs. The polymer sorting (PS) technique is a simple, selective, and scalable process. The polymer molecules are used to wrap up the target SWCNTs in aqueous solution, and then, after centrifugation, the wrapped SWCNTs stay in the solution while unwrapped SWCNTs are moved to sediments. In fact, in toluene, poly(3-dodecylthiophene) (P3DDT) can selectively wrap (7,5) SWCNTs and separate them out with the best efficiency [142]. The density gradient ultracentrifugation (DGU) method is similar to PS because both methods utilize buoyancy force. However, in DGU, different dispersants are selected for correlated SWCNTs to create the density gradient medium. After ultracentrifugation, the SWCNTs are separated into different layers so that high-purity monochiral SWCNTs in aqueous solutions can be obtained [143]. Aqueous two-phase extraction (ATPE) has been demonstrated to isolate semiconducting and metallic SWCNTs via a dextran–polyethylene glycol (PEG) aqueous two-phase system. The mixed constituent phases generate affinity gradients in the aqueous concentrations and separate different SWCNT species due to the different affinities for SWCNTs based on the geometrical structure, surface chemistry properties, and chirality [144].

### 4.1. Devices Based on Randomly Oriented Nanotube Assemblies

There has been much recent interest and progress in fabricating macroscopic CNT assemblies, or architectures, which can be useful for developing scalable quantum technologies. A recent comprehensive review article discusses various potential applications of electroluminescent devices and optical sensors based on CNTs, including macroscopic devices [114]. For instance, Zorn et al. showed EL from ${sp}^{3}$-functionalized CNT networks. The ${sp}^{3}$ defect emission was tunable through the defect concentration and stable over a wide range of current densities, promising electrically pumped single-photon sources in the near-infrared [145].

One established approach to quantum-state generation and information processing is through nonlinear optics based on bulk materials, which offers the generation of entangled photon pairs through parametric down-conversion [146,147]. Compared to bulk nonlinear crystals, CNT films have been demonstrated to produce photon pairs at telecommunication wavelengths of 1.5 μm at room temperature by spontaneous four-wave mixing [148,149]. Such CNT-based devices are promising for future integrated nonlinear quantum devices because they can be made 1000 times thinner than the smallest existing devices, due to the extremely large Kerr nonlinearity of CNTs (≈10^5^ times larger than that of the widely used silica) [148].

As already discussed in Section 3.2.2, formation of polaritons in the SC and USC regimes provides a way to achieve practical devices for QIP at ambient conditions. Polaritons are hybrid light–matter states, and they originate from the coupling between cavity photons and matter in the SC and USC regimes. In particular, exciton-polaritons have been widely studied in SWCNT systems integrated with optical cavities. It is worthwhile to note that exciton-polaritons are not merely a mixture of photons and excitons. They are new quasiparticles with unique properties inherited from both photons and excitons. For instance, they are lighter than electrons because photons are massless. In principle, this allows them to exhibit condensation at room temperature [150]. Moreover, recent developments in exciton-polaritons have led to a growing interest in their potential applications in QIP.

A single exciton-polariton state has been recently probed [151]. Furthermore, polaritonic qubits have been proposed theoretically [152,153]. Tunable SC and USC between SWCNT excitons and cavity photons have been realized at room temperature [154,155,156,157]. Such an SWCNT device provides a way to investigate the application of exciton-polaritons in QIP under ambient conditions. In contrast to the cavity-based devices we described in Section 3.2.2, which were concerned with the interaction between an individual nanotube and cavity photons, all SWCNTs in the film can cooperatively interact with cavity photons to achieve SC because the coupling strength is proportional to the square root of the number of oscillators [158,159,160,161]. In addition, the strong light–matter interaction at room temperature takes advantages of the huge oscillator strengths, high charge mobilities, narrow linewidths, and high exciton binding energies of SWCNTs.

Graf et al. reported SC between SWCNT excitons and microcavity photons, and optically pumped exciton-polaritons at room temperature [157]. A (6,5) SWCNT film was deposited on a thick gold layer above a silicon wafer. A second gold layer was then deposited on top to form a cavity. As shown in Figure 9a, a clear anticrossing pattern and a large Rabi splitting (>110 meV) were observed in the angle-resolved reflectivity data. Cavity thickness dependence further revealed anticrossing (see Figure 9b). The Rabi splitting is cooperatively enhanced by the number of SWCNTs, as can be seen in Figure 9c. It can be estimated from the extrapolation of data in Figure 9c that a Rabi splitting of ∼850 meV can be achieved at 100 wt% SWCNTs, which corresponds to about 70% of the energy of the $E_{11}$ transition, putting the system in the USC regime [157].

Furthermore, in a subsequent study, Graf et al. used the same approach to fabricate a (6,5) SWCNT-based light-emitting field-effect transistor integrated with a microcavity [154], as shown in Figure 9d. Exciton-polaritons with a Rabi splitting of up to 48 meV were observed consistently in angle-resolved reflectivity, PL, and EL measurements at room temperature. The lower polariton occupancy implied that the electrically pumped exciton-polaritons relaxed efficiently towards zero wave vector at high current densities. Figure 9e shows that the coupling strength could be tuned by controlling the gate voltage, due to the unipolar charge carrier accumulation. This allows reversible switching between strong and weak coupling with a Rabi splitting modulation of more than 15 meV [154].

### 4.2. Devices Based on Aligned Nanotube Assemblies

One of the oldest strategies used for fabricating a film of aligned CNTs from a solution sample (after purification) is to utilize a liquid crystal phase transition by mixing SWCNTs with a liquid crystal polymer matrix. The degree of alignment can be extremely high, but the challenge is how to remove such liquid crystal polymers without influencing the alignment of SWCNTs [162]. Another approach is to use self-assembly methods, including evaporation-driven self-assembly, the Langmuir–Blodgett method, and the Langmuir–Shaefer method, which can produce a large-sized and fully covered film without the necessity to remove a polymer wrap. With the Langmuir–Shaefer method, semiconducting nanotubes with purity of 99% have been used to fully cover a surface with a high packing density of approximately 500 tubes per μm [163]. Because of the 2D nature of these techniques, the thickness of fabricated films is limited to a few nanometers.

He et al. developed a new solution-based alignment technique that uses slow vacuum filtration, which can create a uniform, high-density CNT film with precisely controllable thickness (from a few nm to approximately 100 nm) [21]. This process starts from a well-dispersed chirality-enriched CNT aqueous suspension. Then the prepared suspension is poured into the vacuum filtration system. The filter membrane with a pore size smaller than the CNT length blocks the passage of CNTs and allows fluid to pass through. A wafer-scale uniform and aligned CNT film on the filter membrane can be obtained when the following crucial conditions are met: (1) the surfactant concentration must be below the critical micelle concentration; (2) the CNT concentration must be below a threshold value; and (3) the filtration process must be well controlled at a low speed.

Most recently, densely aligned arrays of electronically pure semiconducting CNTs have been realized [23]. Figure 10a–d shows the alignment procedure, which results in wafer-scale nanotube arrays. First, randomly dispersed PCz (poly[9-(1-octylonoyl)-9H-carbazole-2,7-diyl])-wrapped CNTs from the lower solvent start to approach the surface region and become confined in it after 2-butene- 1,4-diol (C_4_H_8_O_2_) is dropped in (see Figure 10a). Next, the wafer is pulled from the solution, and the surface-confined CNTs are left assembled on the wafer due to the strong affinity of C_4_H_8_O_2_ and SiO_2_ (see Figure 10b,c). Finally, a wafer-scale CNT array is formed, as shown in Figure 10d. This method guarantees alignment within 9 degrees and a tunable density between 100 and 200 CNTs per micrometer [23]. Other methods utilized DNA for achieving controlled alignment [24,25]. More specifically, Sun et al. fabricated a CNT array with a precise pitch with three-dimensional DNA nanotrenches [24]. The inter-CNT pitch could be scaled between about 24 and 10 nm. As shown in Figure 10e, the CNTs were confined inside trenches formed between DNA blocks. Figure 10f,g shows transmission electron and atomic force microscope images, indicating the locations of CNTs. The advantages of this method are a uniform pitch, an angular deviation less than 2${}^{\circ }$, and an assembly yield of more than 95% [24].

Aligned assemblies of CNTs have been fabricated not only in film form but also in fiber form. Some of the CNT fibers fabricated to date have an extremely high degree of alignment, a high density, a high aspect ratio (CNT length to diameter) of individual nanotubes, and a low level of impurities [27]. Such CNT fibers have also shown record-high electrical and thermoelectric properties [28,164]. Quantum devices based on CNT fibers are yet to be demonstrated.

Much excitement exists regarding the realization of films of aligned and single-chirality SWCNTs. Such films can be considered to be an extremely anisotropic macroscopic crystal, having periodic potentials in all three dimensions. Efforts in this direction are underway, and there have been a handful of successful studies with promising results for future quantum technology. Gao et al. have shown that the light–matter coupling strength is in the USC regime in a microcavity exciton-polariton system containing a highly aligned single-chirality SWCNT film [156]. An aligned (6,5) SWCNT film was placed at the center of a Fabry–Pérot cavity consisting of PMMA spacer layers and metallic mirrors (see Figure 11b). The PMMA layers were used to tune the thickness of the cavity and adjust the position of the SWCNT film. A very large vacuum Rabi splitting of up to 329 meV was obtained, which was 13.3% of the zero-detuning bare frequency. The anisotropic properties of the SWCNT excitons led to a polarization-dependent vacuum Rabi splitting, enabling the transition from WC to SC and USC regimes, as shown in Figure 11a. Two pairs of exceptional points were observed in the polariton dispersion. The film thickness dependence in Figure 11d confirmed that the coupling strength was proportional to the square root of the number of oscillators, which is a hallmark of Dicke cooperativity [158,159,160,161]. The observations of exciton-polaritons in SWCNT systems at room temperature with a tunable coupling strength pave the way for development of polaritonic applications in QIP under ambient conditions.

## 5. Summary and Outlook

As described in the previous sections, CNTs possess properties that make them promising for applications in future quantum technology. Since their discovery two decades ago, tremendous efforts have been expended to elucidate their 1D electronic, excitonic, and phononic states and behaviors. Recent years have witnessed considerable progress in developing unique CNT devices that take advantage of their quantum properties. Some of the recent highlights in the field of CNT quantum devices include:CNT charge qubits have already demonstrated electric field sensitivity better than that of a single-electron transistor, and DC magnetic field sensitivity comparable to that of NV centers [86].CNT spin qubits with long coherence times and a circuit QED spin–photon interface have been realized [90]. The theoretical limit for the spin coherence time in CNTs can exceed tens of seconds [165]. Moreover, since then, industry has taken over with the goal of realizing CNT spin-based quantum processors [166].Suspended CNTs make perfect nanomechanical resonators with quality factors of up to 5 million [94]. Once a CNT-based nanomechanical qubit is realized, it promises to exhibit a long qubit decoherence time and the capability to couple to a wide range of modalities for external fields [100], which will find applications in quantum sensing and computing.Single-photon emitters hosted in CNTs are especially sought-after in the telecommunication bands as functionalized SWCNTs bring several unique advantages not available in other materials [17].Exciton-polaritons in highly aligned films of SWCNTs form a unique system with on-demand USC [156], which can find applications in quantum information processing and sensing.Due to the electronic and mechanical properties of CNTs, they are used in hybrid quantum devices to improve superconducting circuits [49].

However, some grand challenges that hinder commercial applications of CNTs in quantum technology are yet to be solved, such as: (i) preparation of ultrapure CNTs sorted by chirality and (ii) assembling of CNTs on a macroscopic scale preserving their orientation and separation. More specifically, higher fidelity qubits are yet to be achieved. For spin qubits, in order to remove nuclear spin noise, carbon nanotubes should be grown using purified ^12^C. To further isolate qubits from the environment, suspended CNTs are sought. Another important direction is devising robust schemes for coupling and entangling CNT-based qubits. Regarding CNT-based SPEs, the main challenges are increasing the source brightness, controlling the SPE spectral diversity and generating indistinguishable photons. As discussed in this review, tremendous progress has been achieved in addressing these issues, and the research efforts continue to grow every year. Figure 12 shows the number of publications per year with the keywords “carbon nanotube quantum” (blue bars) and “carbon nanotube” in the Web of Science sorting category “Quantum Science Technology” (orange bars).

In conclusion, we have reviewed several state-of-the-art approaches regarding how CNTs are being investigated and used in devices in quantum technology. We described both devices based on individual nanotubes and devices based on random and aligned assemblies of CNTs. Many degrees of freedom (spin, charge, orbital, and lattice) of CNTs are being utilized for these quantum devices. Thus, CNTs find application in many areas of quantum technology such as quantum computing, communication, and sensing. In addition, we discussed strategies for the preparation of macroscopically aligned assemblies of CNTs. To date, several techniques have been demonstrated to achieve a controlled pitch between nanotubes with high densities [23,24,25] and closely packed nearly perfectly aligned films of nanotubes with tunable thickness [21,22,26]. The next exciting step in this direction is achieving simultaneous macroscopic alignment and chirality sorting to create single-crystal SWCNTs of the same chirality or (*n*,*m*) indices.

We believe that more scientific discoveries and new device realizations for quantum information science and technology based on carbon nanotubes are not far away.

## Figures and Tables

**Figure 1 materials-15-01535-f001:**
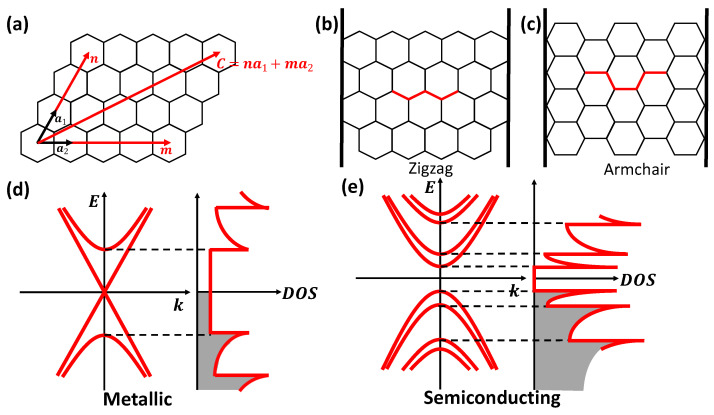
(**a**) Illustration of a graphene sheet in real space, where $a_{1}$ and $a_{2}$ are two primitive lattice vectors and $C=na_{1}+ma_{2}$ defines the roll-up direction of the graphene sheet to produce various helical structures of SWCNTs. (**b**,**c**) Two possible achiral conditions: zigzag and armchair, respectively. (**d**,**e**) Schematics of the band structure of metallic and semiconducting SWCNTs, respectively. Adapted from [14].

**Figure 2 materials-15-01535-f002:**
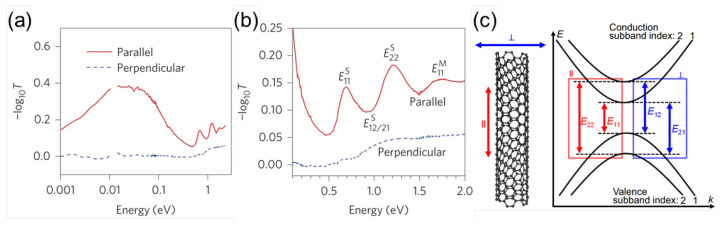
(**a**) Polarization-dependent attenuation spectra in a wide spectral range from the terahertz/far-infrared to the visible. (**b**) Expanded view of (**a**), showing only interband excitonic transitions in the near-infrared and visible range. Adapted from [21] (**c**) Illustration of the lowest-energy allowed optical interband transitions in a semiconducting SWCNT. The numbers shown for the four subbands, two in the conduction band and two in the valence band, are their subband indices. $E_{ij}$ (i=j) denotes an allowed optical transition for parallel (‖) polarization, whereas $E_{ij}$ (i−j=±1) indicates an allowed optical transition for perpendicular (⊥) polarization. Adapted from [63].

**Figure 3 materials-15-01535-f003:**
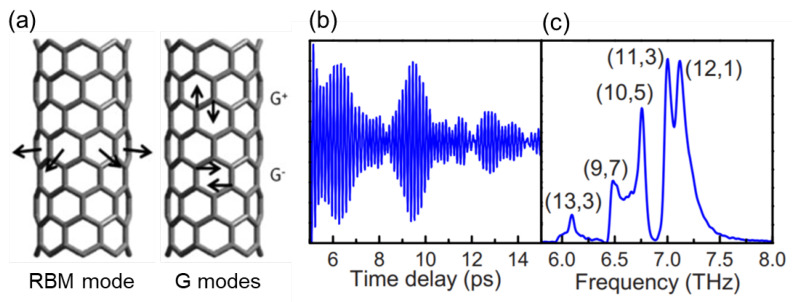
(**a**) Schematic illustrations of vibrational directions for the radial breathing mode (RBM) and the tangential G-modes. Adapted from [77]. (**b**,**c**) Generation and detection of coherent phonons of the radial breathing mode in SWCNTs. (**b**) Time-domain transmission modulations due to coherent RBM vibrations in ensemble SWCNT solution. (**c**) Fourier transform of time-domain oscillations in (**b**) with chirality-assigned peaks. Adapted from [76].

**Figure 4 materials-15-01535-f004:**
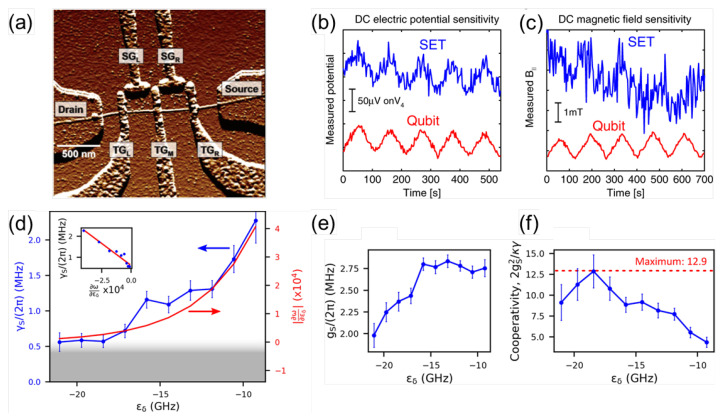
(**a**) AFM image of a CNT double QD device. TG and SG stand for top gate and side gate, respectively. Adapted with permission from Ref. [87]. Copyright 2006 American Chemical Society. (**b**,**c**) Comparison of the performance of a single-electron transistor (SET) and a CNT qubit in sensing DC electric potential and DC magnetic fields. (**b**) DC electric potential measurement. The potential of a single gate (V${}_{4}$) is ramped up and down (saw-tooth) with a period of ∼100 s. (**c**) DC magnetic field measurement. Blue and red traces are for SET and CNT qubit modality measurements, respectively. Adapted from [86]. (**d**) Linewidth and derivative of the dispersion relation of the spin transition as a function of detuning. Inset: linewidth as a function of derivative. (**e**) Spin–photon coupling strength as a function of detuning. (**f**) Spin–photon cooperativity as a function of detuning. Adapted from [90].

**Figure 5 materials-15-01535-f005:**
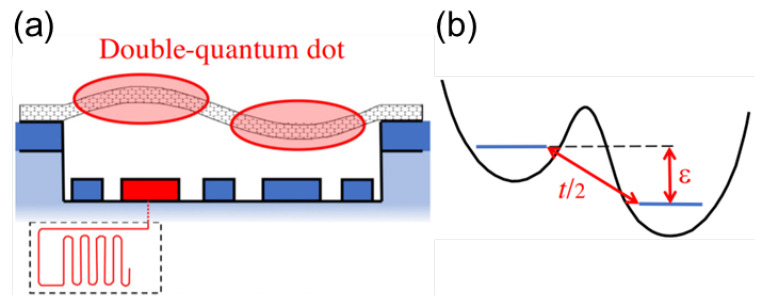
(**a**,**b**) Schematic of the nanomechanical qubit setup. (**a**) A suspended carbon nanotube hosting a double quantum dot, whose one-electron charged state is coupled to the second flexural mode. One of the gate electrodes is connected to a microwave cavity for dispersive qubit readout. (**b**) Sketch of the electronic confinement potential and of the two main parameters, the hopping amplitude *t* and the energy difference ϵ between the two single-charge states. Adapted from [100].

**Figure 6 materials-15-01535-f006:**
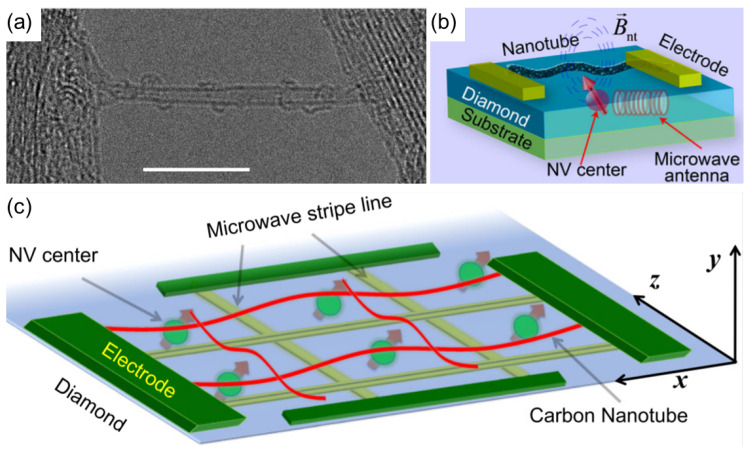
(**a**) High-resolution transmission electron microscopy image showing a single MINT. Four Co-mac-por rings can be seen embracing the SWCNT. Scale bar: 10 nm. Adapted with permission from Ref. [105]. Copyright 2021 American Chemical Society. (**b**) A current-carrying nanotube is suspended above a diamond sample in which individual optically resolvable NV centers are implanted 5–10 nm below its surface [104]. (**c**) Diagram of a quantum hybrid system: a two-dimensional NV centers array in a thin diamond film located under a DC-current-carrying CNT. The array of microwave striplines, which do not cross each other, is used to address the centers independently. Adapted from [109].

**Figure 7 materials-15-01535-f007:**
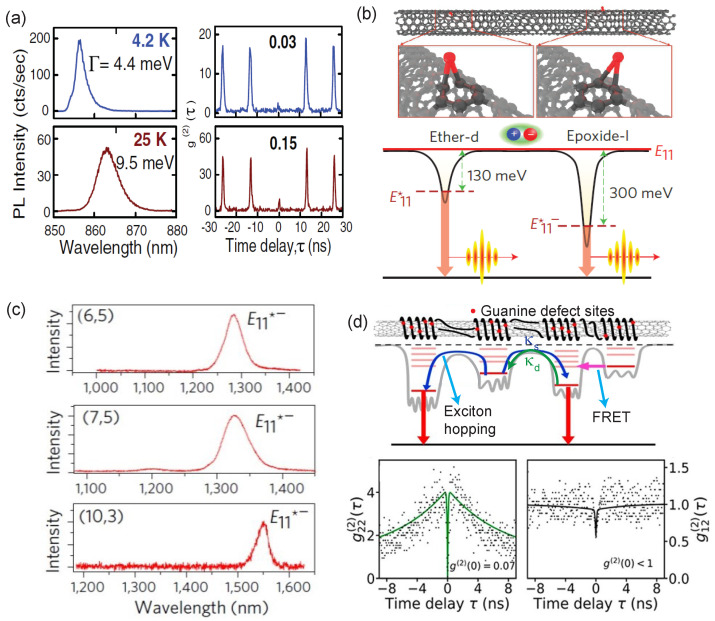
(**a**) Exciton trapping in SWCNTs aiding single-photon emission: temperature-dependent photoluminescence for a single (6,4) nanotube showing decreasing linewidths and vanishing $g^{\left(2\right)}\left(0\right)$ with decreasing temperature. Adapted from [115]. (**b**) Oxygen-doped SWCNTs for room-temperature single-photon emission: creation of deep exciton trap states through introduction of ether-d and epoxide-l groups to the side walls of (6,5) SWCNTs located at 130 meV ($E_{11}^{*}$) and 300 meV ($E_{11}^{*-}$) below the $E_{11}$ state, respectively. Chemical doping of SWCNTs creates trap states with an energy much larger than the $k_{B}T$ value at room temperature, enabling single-photon emission at room temperature. Adapted from [118]. (**c**) The ${sp}^{3}$ defects in various (*n*,*m*) SWCNTs: aryl functionalized oxygen-doped SWCNTs of (6,5), (7,5), and (10,3) chiralities, showing tunable emission wavelengths spanning the O and C telecommunication bands. Adapted from [17]. (**d**) Single-stranded DNA functionalized SWCNTs: schematic showing a nanotube with guanine-defect sites (red dots) on stranded DNA, along with an energy level diagram showing exciton hopping and FRET processes, modeling the cross correlations between the single-photon defect sites. The column figures below represent the auto ($g_{22}^{\left(2\right)}\left(\tau \right)$) and cross ($g_{12}^{\left(2\right)}\left(\tau \right)$) correlation traces for single-nanotube emission. Adapted with permission from Ref. [16]. Copyright 2021 American Chemical Society.

**Figure 8 materials-15-01535-f008:**
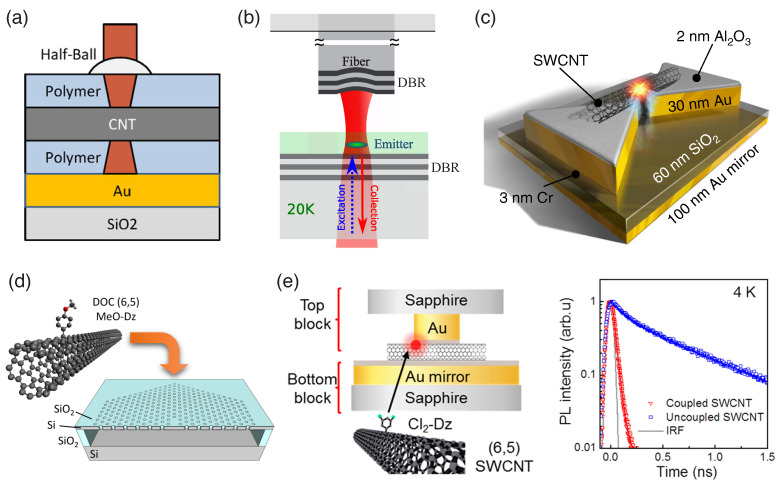
(**a**) Individual SWCNTs inside a cavity consisting of polymer and a metallic mirror. Adapted with permission from Ref. [124]. Copyright 2012 American Chemical Society. (**b**) SWCNT emitter inside a cavity formed by Bragg mirrors on a substrate and at the tip of a flexible optical fiber. DBR: distributed Bragg reflector. Adapted from Ref. [133]. (**c**) Individual nanotube suspended on a plasmonic nanocavity. Adapted from Ref. [132]. (**d**) A doped SWCNT coupled to a two-dimensional photonic crystal microcavity. Adapted from Ref. [134]. (**e**) Left panel: doped SWCNTs coupled to a plasmonic gap-mode nanocavity. Right panel: time-resolved PL for coupled and uncoupled SWCNTs. Blue and red solid lines are fits. IRF: instrument response function. Adapted with permission from Ref. [19]. Copyright 2019 American Chemical Society.

**Figure 9 materials-15-01535-f009:**
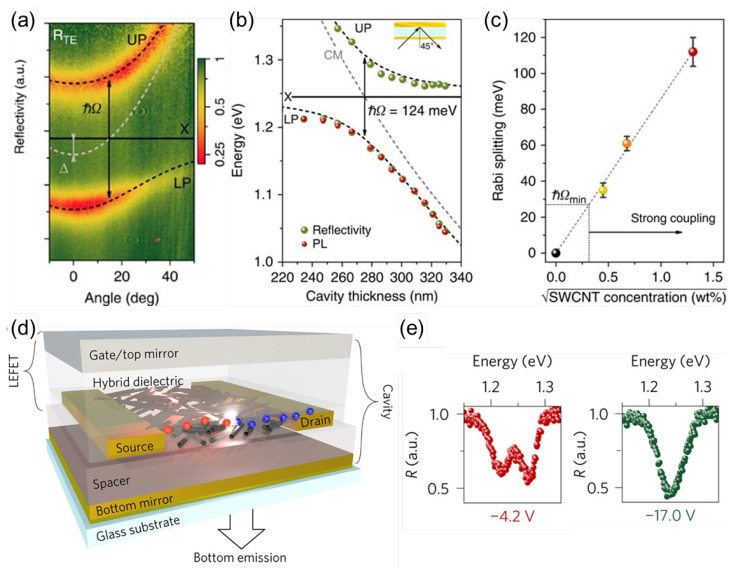
(**a**) Angle-resolved and spectrally resolved reflectivity of a 248 nm thick cavity containing (6,5) SWCNTs. Strong coupling between $E_{11}$ excitons (X, solid black line) and cavity photons (gray dashed line) leads to mode splitting into UP (upper polariton) and LP (lower polariton) branches (black dashed lines, coupled oscillator model fits) with a characteristic Rabi splitting (ℏΩ) of 113 meV. The detuning Δ between the cavity mode and the exciton mode is −17 meV. (**b**) Reflectivity and PL of SWCNT-filled cavity at 45${}^{\circ }$ angle of incidence as a function of cavity thickness. Colored spheres are experimentally determined values for UP and LP from the angle-dependent reflectivity/PL measurements at different cavity thicknesses. The black dashed line shows data from a transfer matrix simulation. The gray dashed line is the cavity mode. (**c**) Experimentally determined Rabi splitting versus the square root of the SWCNT concentration with a linear fit to data. The error bars indicate the standard deviation of the mean. (**a**–**c**) Adapted from [157]. (**d**) Geometry of a bottom-contact/top-gate light-emitting field-effect transistor (LEFET, top stack). By extending the structure with a bottom mirror, an optical microcavity is formed between the top gate and the bottom mirror. (**e**) Reflectivity under a 29${}^{\circ }$ angle of incidence ($E_{X}=E_{C}$) at two different gate voltages. (**d**,**e**) Adapted from [154].

**Figure 10 materials-15-01535-f010:**
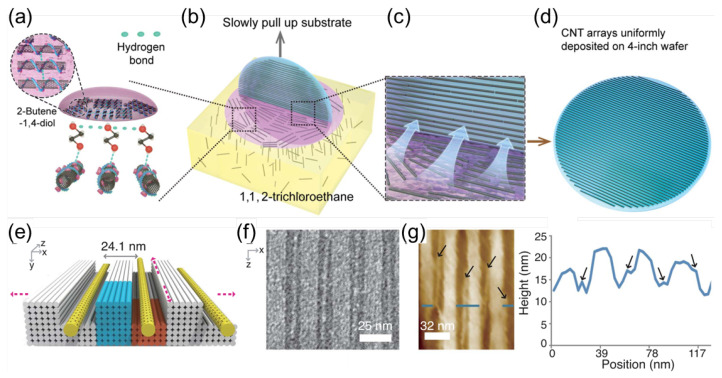
(**a**–**d**) Schematic images showing preparation of a wafer-scale aligned CNT array by a dimension-limited self-alignment procedure. Adapted from [23]. (**e**–**g**): Assembling a CNT array with a 24 nm inter-CNT pitch. (**e**) Design. (**f**) Zoomed-in TEM image along the *x* and *z* projection directions. (**g**) (Left) liquid-mode AFM image along the *x* and *z* projection directions, and (**g**) (right) height profiles of a DNA-assembled CNT array. Adapted from [24].

**Figure 11 materials-15-01535-f011:**
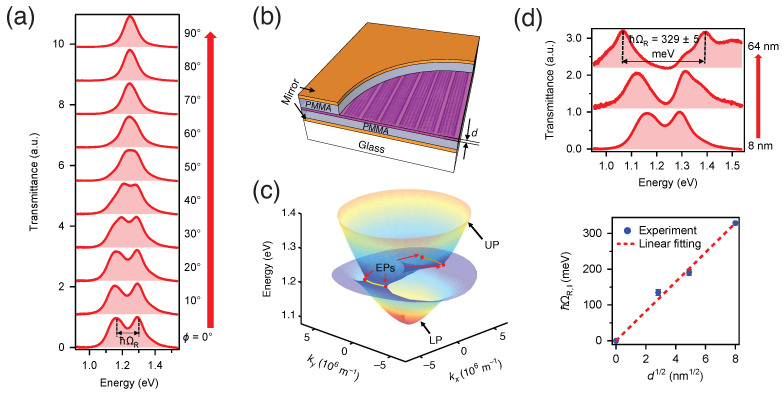
Tunable and ultrastrong light–matter coupling in a microcavity exciton-polariton system based on an aligned and single-chirality SWCNT film. (**a**) Experimental transmittance spectra at zero detuning for various polarization angles ϕ from $0^{\circ }$ to $90^{\circ }$ for a device working in the $E_{11}$ region using an aligned (6,5) SWCNT film with a thickness of 8 nm. (**b**) Schematic of the aligned SWCNT film sandwiched between two metallic mirrors. (**c**) Polariton dispersion surfaces of the SWCNT device. EP denotes exceptional points. (**d**) The top panel demonstrates transmittance spectra at zero detuning for microcavities integrated with SWCNT films of different thicknesses. The bottom panel shows Rabi splitting versus the square root of the film thickness, demonstrating Dicke cooperativity. Adapted from [156].

**Figure 12 materials-15-01535-f012:**
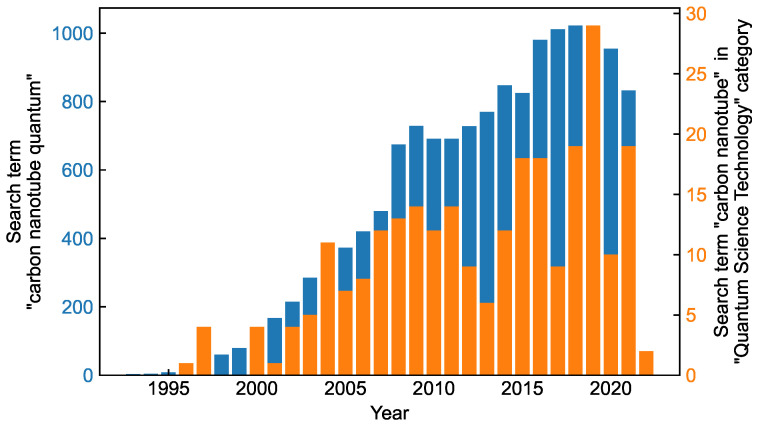
Number of publications per year about quantum applications of carbon nanotubes. Two search terms are shown: (i) “carbon nanotube quantum” (blue) and (ii) “carbon nanotube” in the Web of Science “Quantum Science Technology” category. The data were obtained from Web of Science [167].

## Data Availability

Not applicable.

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
