# Peer review of "Carbon Nanotube Devices for Quantum Technology"

_materials, 2022, doi:10.3390/ma15041535_

Round 1
Reviewer 1 Report
This paper presents an interesting review of the prospects for using carbon nanotubes for quantum applications. This area is new and has a number of technological and fundamental problems that are being actively solved at the moment. The review is well written and reflects current research in the designated area. However, in my opinion, the current aims and problems in the field of creating carbon nanotube devices for quantum technology are not clearly formulated in this paper. In this regard, there are the following comments and recommendations:
- In section 2 “Basics of carbon nanotubes”, it is recommended to add conclusions about the properties and parameters of CNTs (permissible geometric parameters, band gap, defectiveness, etc.), which must be taken into account when developing of quantum devices.
- Figure 1 appears to be redundant in this paper, asreflects the well-known properties of CNTs.In my opinion, a short description in the text is enough.
- There is a typo in the word “ordered” in line 33.
- In section 3. “Devices based on individual carbon nanotubes”, the sentences “CNTs offer a variety of exciting electrical, optical, mechanical, thermal, and magnetic properties.These properties strongly depend on the chirality indices (n,m) that inturn determine the most suitable specific applications, e.g, optoelectronics, bioimaging, and information processing.”(lines 123-126) duplicate the information in section 2. These lines should be deleted.
- The situation is similar for subsection 3.1.Sentences "The 1D nature of CNTs with extraordinary electrical properties showing chirality dependent metallic as well as semiconducting behaviors on the nanoscale makes them one of the most promising materials for replacing copper and silicon in future nanoelectronic, quantum circuits."(lines 135-138) duplicate the information written earlier.I recommend deleting them too.
- Reference 80 (line 147) is not justified, since reference 80 does not deal with CNTs.This statement needs to be edited.
- The statement “…the energy level spacing is large (submillimeter to the THz range)…” (line 150) requires a supporting reference.
- Figure 5a does not carry any useful information and seems redundant.
- Lines 471-474 are written: “Post-processing methods include several purification techniques, which aim is to separate one type of specific chirality of SWCNTs out of a mixture of SWCNTs.The most efficient way to achieve that is to utilize unique properties of SWCNTs like chirality, diameter, length, and electronic type."This is not well written.It turns out that to separate CNTs by chirality, it is necessary to use the unique properties of CNTs, including chirality.This idea should be clarified.
- In section 5. Summary and outlook, it is recommended to describe in more detail the difficulties and the main tasks being solved at the moment in the development of quantum devices based on CNTs, in addition to those indicated in lines 646-647.Since the separation and orientation of CNTs is a common technological problem for all nanoelectronics as a whole.It is necessary to identify frequent tasks and problems in the field of quantum devices.
- The statement “…CNTs, one of the most ideal 1D materials…for applications in future quantum technology” (lines 618-619) is not substantiated, since there is no comparison with other 1D materials in the paper.In addition, the technological realization of quantum devices based on CNTs is currently in doubt, which does not allow us to speak of them as ideal materials.
- The paper should be moved from the Article section to the Review section.
In general, the paper may be accepted for publication in Materials after a revision.
Reviewer 2 Report
The manuscript is nicely written, well design and informative for the readers.
Author Response
We would like to thank the referee for their comments.
Reviewer 3 Report
The authors review recent progress in the development of carbon nanotube-based devices for quantum technology which can revolutionize computation, sensing, and communication. They cover fundamentalproperties of carbon nanotubes and manufacturing methods. Recent developments and proposals for quantum information processing devices, based on individual and assembled nanotubes, are reviewed.
This a systematic and updated review, which can be very useful for the scientific community. I think the article deserves to be published.
Author Response
We would like to thank the referee for their positive and encouraging comments.